# The Role of Structural Variation in Adaptation and Evolution of Yeast and Other Fungi

**DOI:** 10.3390/genes12050699

**Published:** 2021-05-08

**Authors:** Anton Gorkovskiy, Kevin J. Verstrepen

**Affiliations:** 1Laboratory for Genetics and Genomics, Centre of Microbial and Plant Genetics (CMPG), KU Leuven, Gaston Geenslaan 1, 3001 Leuven, Belgium; anton.gorkovskiy@kuleuven.vib.be; 2Laboratory for Systems Biology, VIB—KU Leuven Center for Microbiology, Bio-Incubator, Gaston Geenslaan 1, 3001 Leuven, Belgium

**Keywords:** structural variation, fungi, adaptation

## Abstract

Mutations in DNA can be limited to one or a few nucleotides, or encompass larger deletions, insertions, duplications, inversions and translocations that span long stretches of DNA or even full chromosomes. These so-called structural variations (SVs) can alter the gene copy number, modify open reading frames, change regulatory sequences or chromatin structure and thus result in major phenotypic changes. As some of the best-known examples of SV are linked to severe genetic disorders, this type of mutation has traditionally been regarded as negative and of little importance for adaptive evolution. However, the advent of genomic technologies uncovered the ubiquity of SVs even in healthy organisms. Moreover, experimental evolution studies suggest that SV is an important driver of evolution and adaptation to new environments. Here, we provide an overview of the causes and consequences of SV and their role in adaptation, with specific emphasis on fungi since these have proven to be excellent models to study SV.

## 1. Introduction

Structural variation (SV) groups different forms of mutations that involve longer stretches of DNA, including deletions, insertions, duplications, inversions, translocations, or even full chromosome fusion, fission or loss (Figure 1). Structural variants can be balanced and show no specific loss or gain of DNA information, such as inversions of a genetic fragment or translocations of a stretch of DNA within or between chromosomes, or they can be unbalanced, where a part of the genome is lost (deletions), acquired (insertions) or duplicated (duplications), which is termed copy number variation (CNV). 

Structural variation may occur both in coding and noncoding regions of the genome, including in highly repetitive elements, such as transposons. SV events can lead to major phenotypic changes via diverse mechanisms including modification of open reading frames, changes in gene expression due to copy number variation, alteration of regulatory sequences (via gain or loss of functional genomic elements) or chromatin structure, or even formation of *novel* genes [1,2,3,4,5]. Moreover, some forms of SV, such as large inversions and chromosomal fusions, cause a reduction in recombination rates between homologous chromosome pairs. In turn, the reduced recombination may facilitate the cosegregation of multiple adaptive polymorphisms as if they were controlled by a single genetic locus (linkage disequilibrium and supergene formation) [6,7,8,9,10,11].

In humans, single nucleotide variants (SNVs) are the most common type of variation, but SV accounts for a higher number of variable nucleotides between genomes, with roughly 0.5% of the human genome being involved in structural variation [12,13]. Strikingly, third-generation (long-read) genome sequencing of a clonal population of seven closely related *Schizosaccharomyces pombe* strains that diverged ∼50–65 years ago revealed that they have an average pairwise difference of 19 SNVs and four nonoverlapping larger duplications [14]. Moreover, SVs are three times more likely to be associated with a genome-wide association signal and 50 times more likely to be associated with expressed quantitative trait loci than single nucleotide variants, further hinting at their importance as drivers of phenotypic variation [13,15]. Importantly, despite the significant contribution of SV events (especially of CNVs) to quantitative traits, they are frequently overlooked in studies employing short-read sequencing technologies [14].

The phenotypic consequences of SVs have traditionally been assumed to be almost exclusively negative. This is perhaps partly due to the association of SVs with many human diseases, especially autoimmune, metabolic, and cognitive disorders [16,17,18,19,20]. However, the emergence of advanced genotype-to-phenotype mapping technologies, as well as studies focusing on experimental evolution have led to a growing body of evidence suggesting that many SVs are neutral or even adaptive, both in humans [12,15,21] and other organisms, including microbes [11,22,23,24,25,26,27,28,29,30,31]. SVs are therefore increasingly considered to be an important evolutionary driver, and some studies suggest that SV may be especially important for quick adaptation. 

In this paper, we summarize recent advances in the detection and analysis of SV and the emerging insight into their adaptive role with the focus on yeasts and other fungi, which have served as prime models for many studies focusing on SV.

## 2. Mechanisms of SV Formation

SV involving complete chromosomes is often caused by defective chromosome segregation. Chromosomes must be meticulously replicated and equally segregated at each cell division. Distortion of either one of these processes can lead to SV formation. In particular, failure of any of the critical chromosome segregation steps, including chromatid cohesion, spindle pole body (functional equivalent of the mammalian centrosome) formation at opposite cell poles, kinetochore–microtubule attachment, and quality control at the spindle assembly checkpoint can result in aneuploidy (i.e., loss or gain of whole chromosomes) (Figure 2A) [32].

An SV that does not involve full chromosomes often results from compromised DNA replication, where processive forks collide with the replication fork barriers (Figure 2B) [33,34,35]. These barriers typically include (1) specific DNA secondary structures such as G-quadruplex (G4) motifs [36,37,38], which are enriched in the telomeres, ribosomal DNA (rDNA) and promoter regions in *S. cerevisiae*, *Schizosaccharomyces pombe,* and human cells [39,40,41,42,43,44]; (2) highly expressed loci such as the tRNA genes where transcription can interfere with replication [45,46,47]; or (3) tightly DNA bound nonhistone proteins (e.g., at centromeres) [48,49]. Replication forks can also be stalled as a result of DNA damage or the inhibition of replication by nucleotide depletion [50,51]. Reactivation of blocked replication forks and DNA damage can lead to SV due to the occurrence of nonallelic homologous recombination resulting from incorrect repair template utilization (Figure 2C) [52,53,54]. This process is remarkably more frequent in the case of dispersed repetitive DNA sequences such as transposable elements or remnants of those (long terminal repeats), tRNA genes, origins of replication, and clusters of tandemly repeated genes including those encoding ribosomal RNA and those residing in subtelomeric duplication blocks [14,54,55,56,57,58,59,60,61,62,63,64,65,66,67]. Curiously, stretches of repetitive DNA and, in particular, transposable elements are enriched in highly fast-evolving genomic compartments (which exist as ‘islands’ on core chromosomes) and accessory chromosomes of many pathogenic filamentous fungi [68,69,70,71,72,73,74]. These genomic compartments were shown to be the hot spots of SV [73,74]. Increased plasticity of the indicated genomic regions known to bear the virulence-related genes likely allows pathogens to keep up with the evolution of the host defense mechanisms and succeed in pathogen–host “arms race”.

A third major mechanism underlying SV is linked to crossing-over between repetitive DNA sequences and the repair of DNA double strand breaks near repetitive DNA sequences (Figure 2D,E) [75,76]. Various types of homologous recombination at repeat sites, including unequal crossovers, gene conversion, and single-strand annealing were reported to result in CNV [75]. A specific example of repeat-associated CNV generation, origin-dependent inverted-repeat amplification (Figure 2F), was hypothesized to underlie the amplification of the *SUL1* locus in yeast [77,78]. As a result of the DNA replication error at small, interrupted inverted repeats, nascent leading and lagging strands get covalently linked. This ends up in formation of an extrachromosomal circular intermediate, and its integration into original chromosomal locus results in the gene triplication [77,78]. In some specific cases, copy number amplification is achieved via the formation of the extrachromosomal circular elements [79], which were proposed to be a fast and revocable mechanism of gene copy number amplification [80,81]. 

Finally, a very specific source of gene duplications is the whole genome duplication (WGD, also referred to as polyploidization)—i.e., addition of a complete set of chromosomes to the genome [82,83].

## 3. Gene Duplications: Fruitful Soil for Evolution and Adaptation

The types of SV resulting in a net gain of genetic material such as gene, chromosome segment, full chromosome duplications, as well as WGD events are currently recognized as major sources of adaptation and even evolutionary innovation. Susumo Ohno famously hypothesized that apart from causing differences in gene dosage, gene duplications may create substrates for evolutionary diversity and innovation, including the development of new genes [84,85]. Recent studies focusing on domesticated and experimentally evolved microbes [31,86,87,88,89,90,91,92,93] as well as domesticated higher organisms [94,95] seem to confirm that SV occurs frequently and can in some cases be adaptive. 

The ancestor of the common beer and wine yeast, *S. cerevisiae*, duplicated its genome roughly 100 MYA presumably as a consequence of interspecies hybridization [96]. This whole genome duplication allowed the initially sterile hybrid to restore fertility [97,98,99]. Whereas the *S. cerevisiae* genome contains roughly 5000 genes, the current genome only contains 1120 pairs of duplicates, approximately half of which belong to the WGD event (so-called “ohnologs”) and the remaining are classified as small-scale duplications [100,101]. Thus, the vast majority of genes that were duplicated during the whole genome duplication event eventually returned back to the original copy number [96]. The predominant re-establishment of the original copy number agrees well with Ohno’s classic theory according to which one of the gene copies remains under strong purifying selection while another copy explores novel genotypes, many of which turn out to be detrimental to the organism and are eventually lost [85,102,103]. Both gene copies can be preserved if the resulting increased activity of genes preserving the ancestral function confers an immediate selective benefit (gene dosage effect) [104]. Duplicated genes can also be preserved when they encode different components of a multisubunit protein complex, so that loss of one of the duplicated alleles would cause a stoichiometric imbalance [105,106,107]. In case of multifunctional ancestral genes, subfunctionalization, i.e., the split of the functions between the different gene copies, can drive the conservation of both ohnologs [85,102,103,108,109,110,111]. Lastly, duplicated genes can also be preserved when one copy evolves a novel, beneficial function (neofunctionalization) or a novel expression pattern [112,113,114,115]. Indeed, duplicates were demonstrated to be more transcriptionally polymorphic than singleton genes [116]. Moreover, the fraction of the genes upregulated in stress conditions is significantly higher among ohnologs than among singleton genes in *S. cerevisiae*, suggesting that gene duplication underlies the adaptation to environmental stress [115]. 

Importantly, these different mechanisms that drive the retention of gene duplicates are not mutually exclusive. For instance, dosage constraints were proposed to allow genes to be retained for long enough before changes of function occur [108,117]. Case-by-case variation in the contribution of each particular mechanism is expected. 

## 4. Whole Genome Duplication

Whereas WGD is often considered to be an evolutionary dead end [118], there are many cases of currently polyploid plants, insects, fishes, amphibians, reptiles [119,120,121,122,123] and fungi (reviewed in [124]) which suggests that WGD plays a role in adaptation [125]. Application of the advanced phylogenetic dating methods demonstrated that establishment of polyploids may be promoted during times of environmental cataclysms [118]. In fact, as more and more genomes are sequenced, it becomes increasingly clear that most present-day eukaryotic taxa descended from lineages in which one or multiple WGD events occurred at some point during their evolution [126,127,128,129], including at least two WGDs in the vertebrate lineage [130], and one in *S. cerevisiae* [96,131].

WGD is traditionally classified as either autoduplication or alloduplication depending on whether the doubling of one chromosome set happened within a single species or through the merging of the chromosome sets of different species (hybridization) and subsequent doubling [118]. WGD formation mechanisms include endoreduplication resulting from cytokinesis failure [132] and the loss of function of one copy of the mating-type locus in the hybrid [133,134]. The ancient WGD in the *Saccharomyces* lineage is thought to be a result of alloduplication, involving loss of heterozygosity at the mating-type locus, thereby restoring fertility [97,98]. Both autoduplication and, especially, alloduplication followed by extensive SV events (between and inside of the parental subgenomes) seem to be the major source of genetic diversity and, thus, the propellant of evolution and adaptation in fungi (reviewed in [135]).

The results of various experimental evolution studies in yeast show that the duplication of the genome is a frequent first step towards adaptation. In fact, invasion and subsequent fixation of autodiploids was found to be a common theme in evolving *S. cerevisiae* lineages that started from haploid founder populations [26,93,136,137,138,139,140]. A recent study in *S. cerevisiae* found autodiploids to occur in haploid cultures at a rate on the order of 10^−5^ per cell division [141], similar to that in flowering plants [142], which is much higher than the mutation rates per cell division per base pair of 1.1 × 10^−10^ to 4 × 10^−10^ for haploids [67,143,144] and 1 × 10^−10^ to 3 × 10^−10^ for diploids [143,145,146,147]. Such relatively high WGD rate explains the widely encountered autodiploid invasion in haploid-founded evolution experiments. Consistent with these observations and in accordance with the presumed adaptive potential of WGDs, recent analysis of >1000 yeast genomes revealed that the majority (87%) of natural isolates are diploids, while polyploids (3–5n) are also frequently encountered, especially in specific human-related niches [148]. This can be attributed to the direct fitness advantage conferred by WGD (3.6% fitness benefit in some environments [26]), possibly stemming from the larger cell size of diploid cells [149,150,151] that may facilitate increased growth rates. Moreover, autodiploids were demonstrated to resort to adaptive strategies (in terms of utilized types of mutations) inaccessible to haploids [26]. In particular, aneuploidies and other structural variants accumulate at a significantly higher rate in autodiploids than in haploids, presumably due to buffering of lethal and deleterious recessive mutations [26,143,152,153,154,155,156,157]. For example, Fisher et al. demonstrated that all 46 haploid-founded populations traced in their study converged into diploids by generation 1000. Among these 46 populations, six independently evolved aneuploidies and 20 independently evolved various structural variants [26]. Notably, the spectrum of structural variants generally differs between evolved haploids and diploids. Autodiploid structural variants include both amplifications and (large) deletions, while haploid ones normally result in a net gain of genetic material [143]. 

Whereas diploids are generally more fit than haploids, (fresh artificially created) strains with ploidy >2n tend to be less fit than diploids at least in some growth conditions [158,159]. Additionally, polyploids occur less often in nature compared to diploids [148]. This “ploidy threshold” [160] may at least be partly due to altered geometric relationships between the surface area of the spindle pole body and the length of the preanaphase mitotic spindle, resulting in altered mitotic spindle structure and thus hampered chromosome segregation [159,160]. This agrees with high levels of chromosome loss and interhomolog recombination observed in tetraploid yeasts [161,162,163,164]. Extended propagation of triploid and tetraploid yeasts often results in a gradual loss of chromosomes (aneuploidy) and, eventually, convergence to (near-) diploid state [64,93,140,165]. Moreover, deletion of genes important for accurate chromosome segregation is lethal for tetraploids but not for haploids and diploids, suggesting that proper chromosome segregation is challenging in polyploids [159,166]. However, the increased instability of polyploid genomes may also offer advantages in terms of evolutionary plasticity. Indeed, tetraploids have been shown to boast higher rates of adaptation and higher frequencies and greater diversity of mutations than isogenic haploid and diploid yeast strains that evolved in an environment with a poor carbon source [64,165]. Interestingly, chromosome aneuploidies and/or concerted loss of pairs of chromosomes was observed in 26 out of 28 evolved tetraploid strains at generation 250, suggesting that structural variation could be at the basis of the better adaptation observed in polyploids [64].

## 5. Aneuploidy and Segmental Duplications: A “Quick Fix” for Evolutionary Adaptation?

Aneuploidy is a special form of SV that occurs surprisingly frequently in fungi. Aneuploidy has traditionally been considered as detrimental, in part because in most higher eukaryotes it results in sickness or, more frequently, death. There are multiple reasons why aneuploidy may affect an organism’s fitness, the most obvious one being its effect on gene expression. Doubling the copy number of single chromosomes leads to increased expression of nearly all of the genes on that chromosome, resulting in disproportionate expression of genes residing on different chromosomes [167]. This could be especially harmful for genes encoding subunits of the same protein complex and can result in protein misfolding and aggregation and thus proteotoxic stress. In addition, aneuploidy has also been associated with increased mutation rates, especially on the duplicated chromosome(s), and chromosome mis-segregation [168,169,170,171,172,173,174]. Moreover, overabundant protein subunits increase intracellular solute concentration, resulting in high cytoplasmic osmolarity in aneuploid cells. As a result, the aneuploidy-associated transcriptome signature common for aneuploid cells with random karyotypes closely resembles that of cells experiencing osmotic stress [175]. In the *S. cerevisiae* laboratory strain W303, aneuploidy also leads to severe growth defects, metabolic dysfunction, transcriptome changes and cell-cycle defects [167,169,176,177,178,179,180,181]. 

Despite the multitude of studies that report the detrimental effects of aneuploidy, many yeast strains coming from diverse ecological niches, including many clinical isolates and industrial strains, are aneuploid [86,182,183,184,185,186,187,188,189]. It was demonstrated that 19.1% of natural *S. cerevisiae* isolates contain chromosome aneuploidies and instances of large segmental duplications, especially for chromosomes I, III and IX [148]. Interestingly, only mild growth reduction was observed in natural aneuploid *S. cerevisiae* yeast strains compared to (artificially generated) isogenic euploid counterparts, and there did not seem to be clear evidence of metabolic or proteotoxic stress in tested growth conditions (on media having glucose, ethanol, glycerol, and acetate as a carbon source) [182,190]. The difference in the tolerance for aneuploidy between the highly intolerant *S. cerevisiae* strain W303 and more tolerant wild strains was recently mapped to the RNA-binding translational regulator Ssd1, which is defective in W303 [191]. Ssd1 is involved in regulating where, when, and to which extent mRNAs are translated, which is essential for enabling the cells to minimize protein aggregation and misfolding and, thus, to cope with the burden imposed by extra chromosomes. Consistently, in another laboratory strain, S288C, bearing a full-length copy of the *SSD1* gene, aneuploidy was shown not to result in proteotoxic stress [192]. 

More generally, the results of the recent study of over 1000 published *S. cerevisiae* genomes suggest that genetic background influences variation in aneuploidy frequency and cellular tolerance of aneuploidy stress [193]. It is likely that aneuploidies are near-neutral in many strains and, once they appear, may persist in the population [193]. Furthermore, it is more and more evident that aneuploidy can actually also be beneficial for adaptation, especially when gene expression issues are resolved [67]. Indeed, aneuploidies were shown to drive the adaptation to different types of environmental stress, including limiting nutrients, high ethanol concentration, heat shock, oxidative stress, endoplasmic reticulum stress, and drug resistance [31,92,93,155,194,195,196,197,198,199,200,201]. In addition, aneuploidy has been shown to help suppress certain genetic mutations such as telomerase insufficiency [202,203], cytokinesis perturbation [91], disruption of essential nucleoporin genes [204] and loss of a small ubiquitin-related modifier protease [205]. In many cases, the selective advantage can be attributed to copy number changes of a few genes on the aneuploid chromosomes [91,196,199,205,206]. In addition, several studies indicate that aneuploidy leads to increased spontaneous mutation rates [67,168,173], which may of course also result in beneficial mutations that can drive adaptation. Interestingly, fixed mutations in disomic lines are frequently located on the duplicated chromosomes, likely due to relaxed purifying selection on the duplicated genes [31,67]. 

Aneuploidies appear in *S. cerevisiae* diploid cells with a frequency close to that of WGD events (10^−5^ to 3 × 10^−4^ per cell division); generally, chromosome gains prevail over the chromosome losses [143,146,207]. As mentioned above, the frequency of aneuploidy is further increased in cells with >2n ploidy [64]. 

Whereas aneuploidy can help improve fitness by changing the expression of certain genes, the possible advantage is often upset by the general disadvantage of expression imbalances for many other genes. Hence, aneuploidy is often not a very stable state, with subsequent genomic rearrangements gradually reducing the negative effects. The cellular response to aneuploidy has been analyzed at both the transcriptome and proteome levels in *S. cerevisiae* [67,90,175,179]. In disomic strains, the aneuploidy response involves loss of parts of the duplicated chromosomes, with a clear tendency towards euploidy restoration. In addition to reducing the copy number of certain duplicated segments, the opposite has also been observed. Segmental duplication of nondisomic chromosomes was observed in response to duplication of chromosomes XII and XIV, which could balance out the overexpression of the (particular) genes on the disomic chromosomes [67]. This is in line with the results of Ravichandran et al., who observed that yeast stains engineered to have high rate of chromosomal mis-segregation due to *BIR1* deletion, in time restored their fitness via acquisition of complex karyotypes that consisted of specific subsets of the beneficial aneuploid chromosomes [174]. The “optimal” aneuploid karyotypes shaped via the gain and loss of chromosomes were dictated by genetic interactions between aneuploid chromosomes. For instance, in diploids, loss of chromosome IX strongly correlated with the gain of chromosomes X or XIII, while in haploids a significant negative correlation between the gain of chromosomes VIII and X was observed [174]. 

Interestingly, the stress caused by aneuploidy is not only resolved by further changes in the copy number of certain genomic segments, but can also be mitigated by compensatory mutations. For example, in *S. cerevisiae*, aneuploidy may select mutations in the *SCH9* kinase gene. These mutations may reduce *SCH9*-mediated ribosome biogenesis and thus decrease translation initiation, which may help in reducing the proteotoxic stress associated with an increased gene copy number [67]. Similarly, variation in *SCH9* has also been implicated in the stabilization of tetraploidy [158].

Aneuploidy is often hypothesized to be a resourceful adaptive mechanism that allows a “quick fix” solution to immediate threats by changing the expression of certain key genes. While this may also bring undesirable changes in the expression of other genes, it may allow cells to quickly adapt to the stress, after which mutations and additional SVs may result in a gradual further tuning of the genome [157,208,209]. Hence, aneuploidy may be able to fuel fast phenotypic evolution and allow cell populations to rapidly explore adaptive mechanisms, eventually leading to large fitness gains [64,210]. 

Akin to aneuploidy, other types of unbalanced SV events, such as segmental duplications and deletions, may also allow fast adaptation to new environments, stress, deleterious mutations or adaptive mutations that have trade-offs. In *S. cerevisiae*, the estimates of the frequency of such events differ drastically, ranging from 1 × 10^−10^ to 1 × 10^−4^ duplications per cell per division, suggesting that the rates may depend on the genomic context of the locus of interest (e.g., the existence of direct repeats flanking the locus, the distance between the repeats, the sequence similarity between the repeats), ploidy level, growth conditions and the genetic background of the organism, which may affect its ability to tolerate the CNV event [211,212,213,214,215]. Indeed, segmental duplications and deletions are not evenly distributed across the yeast genome. Eukaryotic subtelomeres are known hot spots of these events [216,217,218,219]. According to the results of the recent genomic study of industrial *S. cerevisiae* strains, the subtelomeric regions of chromosomes, defined as gene-depleted regions adjacent to telomeres [220,221], are on average four times more frequently affected by CNV events compared to nonsubtelomeric regions, with most variability detected in subtelomeres of chromosomes I, VII, VIII, IX, X, XII, and XVI [86]. In another study, alternative definition of subtelomeres on the basis of the sudden loss of synteny conservation was proposed [65]. Analysis of long-read genome assemblies of *S. cerevisiae* S288C standard laboratory strain showed that most previously described duplication blocks are located in subtelomeres [65]. Since many subtelomeric genes are known to mediate the interaction of yeast with the environment, including functions such as stress response, nutrient uptake and ion transport [87,221,222], it was hypothesized that the accelerated evolution of subtelomeric regions echoes the selection for evolvability—i.e., the ability to respond and adapt to changing environments [65,223]. 

## 6. Balanced SV Events

Balanced SV types such as reciprocal translocations and inversions are widespread in *Saccharomyces* species and other fungi [14,60,63,65,71,72,224,225,226,227,228,229]. They are thought to serve as initial genetic barriers in eukaryotic speciation and, thus, to contribute to the onset of reproductive isolation and speciation [230,231,232,233,234,235,236]. In flies [237], mosquitoes [238], and flowering plants [8], inversions are hypothesized to also play a role in evolutionary adaptation. Analysis of the outcomes of chromosomal translocations in *S. cerevisiae* [239] and of translocations and inversions in *S. pombe* [225,240] demonstrated that these types of SV can significantly influence the fitness of the organism in specific environments, possibly as some events cause changes in gene expression [225]. It was hypothesized that balanced types of SV can be maintained as polymorphisms in nature despite their meiotic costs (low viability in heterozygotic crosses) when this disadvantage is outweighed by the fitness advantage gained in mitosis (antagonistic pleiotropy) [225]. Contrastingly, Naseeb and colleagues were not able to detect phenotypic consequences of a set of large inversions, even if they did observe significant changes in gene transcription patterns [241]. This again underscores that the effect of a specific structural rearrangement always depends on the affected genetic locus, the genetic background and the environment.

## 7. Examples of Adaptive SV Events

As the number of sequenced genomes and experimental evolution studies is growing, there is an increasing number of examples of evolutionary adaptation driven by SV events (Table 1). In a subset of cases, the particular gene or genes underlying the adaptation have been revealed. In most cases, gene dosage of these genes played a major role, which agrees with the fact that SV events detected in *S. cerevisiae* are predominantly unbalanced events, such as insertions, deletions and duplications [65]. Among the genes that are gained by *S. cerevisiae* after the WGD event, many have functions associated with ethanol production, growth in hypoxic environments or the uptake of alternative nutrient sources [63,242,243]. The increased copy number of glycolytic genes and glucose transporters originating from WGD is thought be responsible for the boosting of the glycolytic flux and allow the adaptation to niches with high levels of sugars, including fruits and industrial media [100,244,245]. CNVs of nutrient transporter genes were also repeatedly detected in strains evolved to adapt to various nutrient-restricted conditions such as glucose limitation (high-affinity glucose transporters *HXT6* and *HXT7*) [31,246], sulfate limitation (high-affinity sulfate transporter *SUL1*) [30,196], media containing poor nitrogen sources such as glutamine or glutamate (general amino acid permease *GAP1*) [79,247], allantoin (permease *DAL4*) or urea (permease *DUR3*) [139]. Interestingly, different classes of CNVs, including aneuploidies, nonreciprocal translocations, tandem duplications, and complex CNVs, that contain the general amino acid permease *GAP1* were shown to be repeatedly generated and selected upon growth in nitrogen-limiting conditions [247]. Interestingly, in addition to affecting copy numbers, nonreciprocal translocations were also shown to result in increased expression levels of genes located near the CNV breakpoints, presumably due to a more open state of their chromatin structures, allowing easier access of the DNA by the transcription machinery [248]. 

The recent genome-wide association study performed on 1011 yeast strains identified 22 CNVs that were strongly associated with improved growth under stress conditions such as at high temperature (40 °C, *PAU5* and several genes introgressed from *S. paradoxus*), elevated concentration of copper sulfate (metallothionein *CUP1*), sodium chloride (Na^+^ efflux ATPase *ENA5*, component of the mitochondria-ER-cortex-anchor *MDM36* and *VAR1589* introgressed from *S. paradoxus*), lithium chloride (*ENA5* and *VAR1589*), antifungal drug nystatin treatments (alcohol dehydrogenase *ADH4* and hexokinase *HXK2*), presence of sodium meta-arsenite (subtelomeric region of chromosome XVI containing *ARR1*, *ARR2*, and *ARR3* genes, known be essential for resistance to arsenic compounds [280] as well as *YPR196W*, *YPR195C*, *SGE1*, *AQY1*, and *HPA2*), and on medium with galactose as the carbon source (two genes introgressed from *S. paradoxus*) [148]. 

Importantly, these associations between specific CNVs and stress tolerance are often confirmed by other studies. The results of the recent third-generation sequencing effort that provided a high-resolution picture of the *S. cerevisiae* and *S. paradoxus* subtelomeric regions, and *CUP1* and *ARR* clusters in particular, further confirm these associations between the gene copy numbers in these clusters and resistance to high-copper and high-arsenic conditions [65]. Similarly, the duplication of *CUP1*, alongside with increased copy number of *CUP2*, *SCO1*, and *SCO2*, were previously implicated in increased copper tolerance of both environmental *S. cerevisiae* isolates and strains subjected to experimental evolution [250,251,252]. Another example of CNV-mediated stress adaptation is freeze-thaw tolerance of an environmental isolates of *S. cerevisiae* and *Saccharomyces paradoxus*, which was associated with CNV involving the *AQY2* water-transporter gene [182,253]. Increased expression of 17 genes due to duplication of chromosome III (of which *HCM1*, *YCR016W*, *RRT12*, *YCR102C*, and *IMG2* showed the highest contribution) was linked to increased heat tolerance of experimentally evolved *S. cerevisiae* strains [195]. Large segmental duplications and aneuploidization involving this chromosome were iteratively found in association with improved heat tolerance [254,255]. 

Extensive genomic and phenomic studies revealed that CNVs also underlie some of the specific characteristics of industrial *S. cerevisiae* strains [86,87,89]. The genes most frequently present in variable copy numbers in different strains include those involved in nitrogen and carbon metabolism, ion transport, and flocculation [86,87,89]. Interestingly, there seem to be an association between some CNVs and particular environmental niches, indicating the potential adaptive nature of those CNVs [86]. For instance, genes involved in uptake and breakdown of maltose, such as *MAL1*, *MAL3*, *MPH2, MPH3*, and *YPR196W*, are often amplified in beer strains that are adapted to fermenting maltose-containing wort, whereas they are often lost in wine strains that are adapted to grape must, which does not contain maltose [86,271]. In addition, CNVs involving the flocculation gene *FLO1, Lg-FLO1, FLO5 and FLO10* were implicated in conferring desirable cell aggregation in industrial beer strains [184,272]. Low diacetyl production, another industrially relevant characteristic of *S. pastorianus* beer strains, was shown to rely on increased copy number of *ILV5* and *ILV3* genes, encoding enzymes that catalyze reactions converting diacetyl precursor α-acetolactate into α-keto-isovalerate, thus reducing the amount of α-acetolactate that can be converted into diacetyl via chemical oxidative decarboxylation [184].

In wine *S. cerevisiae* isolates, potentially adaptive CNVs include loci containing genes encoding various transporters, dehydrogenases and genes involved in the metabolism and efflux of toxic subtances that are sometimes used in winemaking, such as copper and sulfite [88,281,282]. Similarly, for *S. cerevisiae* dairy strains, duplication of the galactose permease *GAL2* gene and the introgressed *GAL7-GAL10-GAL1* gene cluster allows elevated galactose utilization rate [187]. Another example of domestication-driven phenotype boosted by gene duplication is the copy number amplification of α-amylase gene in filamentous fungus *Aspergillus oryzae* used for metabolizing starch found in rice kernels [273,274]. 

SV events leading to increased copy numbers of certain genetic loci were shown to be involved in increasing drug resistance of various fungi. The recurrent theme is that increasing the dosage of the drug targets or of genes that help to clear the drugs, convey resistance. For instance, resistance towards the widely used antifungal drug fluconazole in a broad range of yeasts, ranging from *S. cerevisiae* laboratory strains and clinical isolates to pathogenic *Candida albicans* and *Cryptococcus neoformans* strains, is acquired via gain of extra copies of *ERG11*, encoding the lanosterol 14-alpha-demethylase enzyme in the ergosterol pathway that is targeted by this drug [194,262,263,264,265,266]. In addition, duplication of *TAC1*, *MRR1*, *CDR1* and *CDR2* in *C. albicans* and *AFR1*, *SEY1* and *GLO3* in *C. neoformans* were also reported to be involved in conveying resistance to fluconazole [262,263,264,265,266]. Resistance to another antifungal drug, itraconazole, in the human pathogen *Aspergillus fumigatus is conferred by the* extra copies of cytochrome P-450-depdendent C-14 lanosterol α-demethylase *pdmA* [268]. Increased copy number of the mutated allele of *CYP51* gene is associated with the increased resistance towards sterol demethylase inhibitor fungicides in grape powdery mildew pathogen *Erysiphe necator* [269]. *Similarly, in*
*S. cerevisiae*, chromosome XV aneuploidy-mediated increase in gene dosages of Hsp90 cochaperone *STI1* and multidrug transporter *PDR5* drives the resistance to Hsp90 inhibitor radicicol [194], while resistance to the endoplasmic stress inducer tunicamycin can be achieved by duplication of several genes located on chromosome II, including UDP-N-acetylglucosamine-1-P transferase *ALG7*, a subunit of the 20S proteasome *PRE7*, and *YBR085C-A* [199]. Finally, resistance to the tumorigenic compound 4-nitroquinoline-N-oxide is conferred by an extra copy of the multidrug efflux pump *ATR1* [90]. 

The effects of some deleterious mutations can also be suppressed by CNV events. In *S. cerevisiae,* for example, the deletion of the essential genome integrity checkpoint gene *MEC1* is rescued via duplication of large subunit of ribonucleotide-diphosphate reductase *RNR1* [276]. Similarly, in diploids, telomerase insufficiency triggered by growth at elevated temperatures and associated with the deficiency of telomerase catalytic subunit Est2 [202] is suppressed by chromosome VIII monosomy, resulting in the reduction in the copy numbers of *PRP8*, *UTP9*, *KOG1*, and *SCH9* genes that are connected to ribosome production [203]. The slow growth phenotype resulting from the deletion of *RNR1* and *RPS24A* can be rescued by increased gene dosage of the paralogues of the deleted genes (*RNR3* and *RPS24B*, respectively) [206]. Deletion of the type II myosin heavy chain gene *MYO2* involved in cytokinesis is suppressed by the duplication of transcription factor *RLM1* implicated in cell wall remodeling and MAP kinase kinase *MKK2* regulated by the indicated transcription factor [91]. The laboratory evolution of *S. cerevisiae* strain bearing the deletion of the only documented lactate transporter *JEN1* for restoration of lactate transport resulted in duplication of the *ADY2* monocarboxylate-transporter gene and was accompanied by point mutations of the duplicated allele, increasing its lactate transporter efficiency [277]. In a *S. cerevisiae* model of galactosemia, a *Δgal7* strain characterized by the buildup of galactitol and D-galactose-1P, an extra copy of the transcriptional repressor of multiple genes in the galactose utilization pathway *GAL80* was shown to mediate the galactose tolerance [278]. In *C. albicans*, the fluconazole resistance lost after deletion of RHO-GTPase activator gene *RGD1* was restored via duplication of a putative urea *transporter NPR2* [279]. 

There are also many instances when the aneuploidies were shown to be associated with the beneficial phenotypes, but in these instances, pinpointing the loci that drive the adaptation is often difficult. In *S. cerevisiae*, chromosome III polysomy was identified in the highest ethanol-tolerant natural and fermentative strains [256]. Duplication of the same chromosome as well as chromosome XII was observed in ethanol-tolerant variants resulting from a directed evolution study where populations were grown in the presence of high ethanol concentrations [93]. Duplication of chromosome XI is associated with increased ethanol yield [28]. Presence of an extra copy of chromosome XIV was linked to improved xylose fermentation in the presence of ferulic and p-coumaric acids [275]. Acquisition of an additional copy of chromosome XIII improved growth of tetraploid *S. cerevisiae* strains in a poor carbon-source medium raffinose [64]. Interestingly, the effect seemed exclusive to tetraploids and was not observed in diploids. 

Akin to CNV events, aneuploidies can also affect the pathogenicity of yeasts. In *S. cerevisiae* clinical isolates, aneuploidies have been linked to increased host survival [185]. During experimental oropharyngeal infection in mice, *C. albicans strains bearing* triplication of chromosomes V or VI were able to achieve the same oral fungal burden as the diploid progenitor strain while eliciting a significantly lower inflammatory host response and causing significantly less weight loss of the infected animals (commensal-like phenotype) [270]. The loss of chromosome V increased tolerance to antifungals with different mechanisms of action (fluconazole, andamphotericin B, and caspofungin) [200,267], as well as conveyed the ability to utilize *L*-sorbose as a carbon source due to upregulation of sorbose *SOU1* [249]. Finally, chromosome II trisomy was shown to increase *C. albicans* tolerance to hydroxyurea or caspofungin [200].

Interestingly, there are also examples of adaptation mediated by SV events that do not involve gene duplication. For example, inversion of the *DAL2* gene encoding allantoicase in S. cerevisiae results in decreased expression of this gene and reduced yeast fitness during nitrogen starvation. This SV event also changed the expression of the neighboring genes *DAL1* and *DAL4* [283]. Among wine yeasts, adaptive reciprocal translocations between chromosomes VIII and XVI (VIII-t-XVI) and between chromosomes XV and XVI (XV-t-XVI) are widespread [29,257,258,259,260]. Both translocations result in shortened lag phase in a medium containing sulfite, a commonly used additive that prevents wines from oxidation. The effect of these translocations was traced down to the increased expression of plasma membrane sulfite pump *SSU1* due to a promoter switch mediated by the microhomology between the promoters *SSU1* and *ECM34* (in case of VIII-t-XVI) and *SSU1* and *ADH1* (in case of XV-t-XVI) [29,258,260]. Intriguingly, engineering of the translocations present in *Saccharomyces*
*mikatae* isolated from nutritionally poor habitats into *S. cerevisiae* allowed the resulting strains to outcompete the parent strain under different physiological conditions, especially under glucose limitation [239,284]. Moreover, an inversion in chromosome XVI mediated by a microhomology between the *SSU1* and *GCR1* regulatory regions was also shown to increases *SSU1* expression and thus sulfite resistance [261]. 

Another (quite striking) example of the adaptation associated with balanced SV events (translocations and, to a lesser extent, inversions) is the modulation of the aggressiveness towards host of the fungal plant pathogen *Verticillium dahliae* [72]. By combining short-read and mate-pair sequencing with optical mapping, De Jonge and coauthors discovered that the genomes of the highly and mildly virulent strains of *V. dahliae* were 99.98% identical in all genomic regions that could be aligned differ by 11 intra- and 17 interchromosomal rearrangements [72]. These SV events shaped the lineage-specific segments that contain the genes associated with pathogenicity (*Ave1, XLOC_009059*, *XLOC_008951* and others) [72].

Finally, with the ever-increasing molecular toolbox, it has recently become possible to induce SV at an unprecedented scale and thus investigate the phenotypic outcomes of such induced SV events. One of the most striking demonstrations of the adaptive power of SV has come from the ongoing Sc2.0 project where a large team of researchers is assembling a synthetic *S. cerevisiae* genome interspersed with recombination sites that can be triggered to induce SV. Such “reshuffling” of chromosomes (synthetic chromosome rearrangement and modification by loxP-mediated evolution—SCRaMbLE [285]) resulted in multiple SV events per strain and lead to large phenotypic diversity in a wide range of growth conditions allowing identification of strains adapted to high alkali conditions [286], caffeine [287], heat [287,288], ethanol [288], or acetic acid [288]. Additionally, SCRaMbLE has been used to boost the effectiveness of yeast in production of various heterologous metabolites including violacein, carotene, lycopene, and betulinic acid [287,289,290,291,292,293]. 

## 8. Conclusions

A growing body of evidence reveals the important role of SV in fungal evolution and adaptation. Whereas the majority of examples of adaptive structural variation correspond to CNVs, others can be attributed to transcriptional changes of the genes located within or near the SV event. Aneuploidies as well as other unbalanced SV events resulting in net gain of the genetic material, especially ones affecting the subtelomeric regions characterized by the rampant reshuffling of the genes, seem to serve as an evolutionary “quick fix” solution, allowing yeast populations to rapidly adapt to acute stress, after which further SVs and other mutations gradually compensate the potential detrimental side effects of the original SV event. However, while the appreciation for the adaptive role of SV is rising, many SV events are currently still overlooked due to technical challenges in their detection. The emergence of the third-generation long-read sequencing technologies [294] and next-generation mapping technologies, such as optical mapping, 10x Genomics linked reads and chromosome conformation capture techniques such as Hi-C, will likely uncover many more SV events and put SV even more centerstage in molecular evolution, where it may finally claim its spot next to the more established role of SNPs and smaller InDels.

## Figures and Tables

**Figure 1 genes-12-00699-f001:**
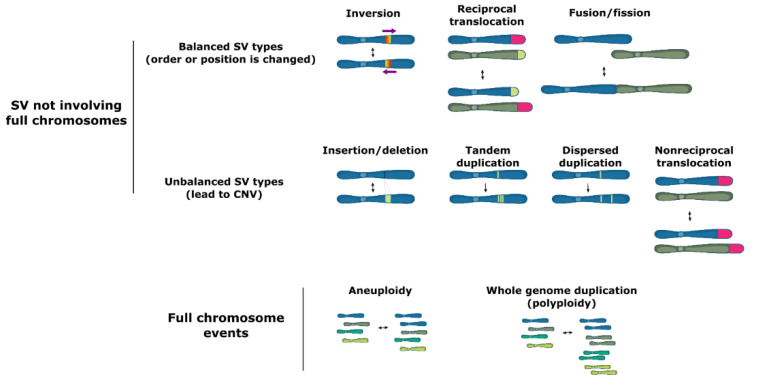
Types of structural variation.

**Figure 2 genes-12-00699-f002:**
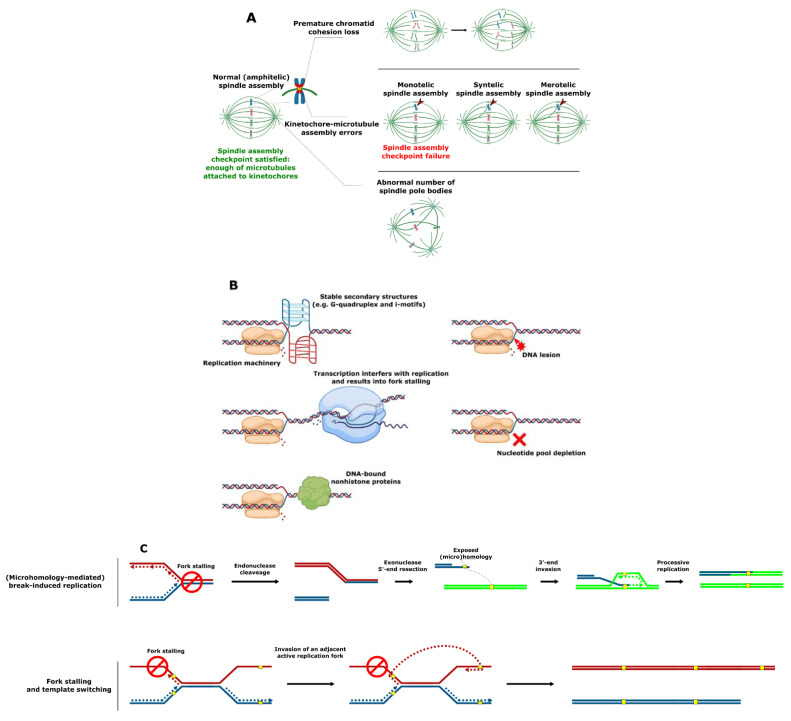
Mechanisms of SV formation. (**A**) Events leading to aneuploidy. (**B**) Events leading to replication fork collapse. (**C**) SV formation as a result of stalled replication fork reactivation. (**D**) SV formation mediated by homologous recombination. (**E**) SV formation mediated by nonhomologous end joining. (**F**) Origin-dependent inverted-repeat amplification.

**Table 1 genes-12-00699-t001:** Examples of adaptive SV events.

Phenotype	Gene(s)	Type of SV	Organism	References
Adaptation to:				
Glucose limitation	*HXT6* and *HXT7*	Increased copy number	*S. cerevisiae*	[31,246]
Sulfate limitation	*SUL1*	Increased copy number	*S. cerevisiae*	[30,196]
Poor nitrogen sources (glutamine or glutamate)	*GAP1*	Aneuploidies, nonreciprocal translocations, tandem duplication, complex CNVs	*S. cerevisiae*	[79,247]
Poor nitrogen sources (allantoin)	*DAL4*	Increased copy number	*S. cerevisiae*	[139]
Poor nitrogen sources (urea)	*DUR3*	Increased copy number	*S. cerevisiae*	[139]
Poor carbon sources (raffinose) ^1^		Chr XIII duplication	*S. cerevisiae*	[64]
Poor carbon sources (*L*-sorbose)	*SOU1*	Chr V monosomy	*C. albicans*	[249]
High temperature (40 °C)	*PAU5* and several genes introgressed from *S. paradoxus*	Increased copy number	*S. cerevisiae*	[148]
Elevated concentration of copper sulfate	*CUP1*, *CUP2*, *SCO1*, and *SCO2*	Increased copy number	*S. cerevisiae*	[65,148,250,251,252]
Elevated concentration of sodium chloride	*ENA5*, *MDM36*, and *VAR1589* introgressed from *S. paradoxus*	Increased copy number	*S. cerevisiae*	[148]
Elevated concentration of lithium chloride	*ENA5* and *VAR1589*	Increased copy number	*S. cerevisiae*	[148]
Sodium meta-arsenite	Chr XVI subtelomeric region containing *ARR1*, *ARR2*, and *ARR3* genes	Increased copy number	*S. cerevisiae*	[65,148]
Nonpreferred carbon sources (galactose)	Two genes introgressed from *S. paradoxus*	Increased copy number	*S. cerevisiae*	[148]
Freeze–thaw cycles	*AQY2*	Increased copy number	*S. cerevisiae S. paradoxus*	[182,253]
High temperature (39 °C)	*HCM1*, *YCR016W*, *RRT12*, *YCR102C*, and *IMG2*	Increased copy number	*S. cerevisiae*	[195]
High temperature (42 °C)		Chr III segmental duplication and aneuploidy	*S. cerevisiae*	[254,255]
High ethanol concentration		Chr III and Chr XII duplication	*S. cerevisiae*	[93,256]
Sulfite	*SSU1*	Translocations VIII-t-XVI XV-t-XVI	*S. cerevisiae*	[29,257,258,259,260]
Sulfite	*SSU1*	Inversion in Chr XVI	*S. cerevisiae*	[261]
Nystatin	*ADH4* and *HXK2*	Increased copy number	*S. cerevisiae*	[148]
Fluconazole	*ERG11*	Increased copy number	*S. cerevisiae* *C. albicans* *C. neoformans*	[194,262,263,264,265,266]
Fluconazole	*TAC1*, *MRR1*, *CDR1* and *CDR2*	Increased copy number	*C. albicans*	[262,266]
Fluconazole	*AFR1*, *SEY1* and *GLO3*	Increased copy number	*C. neoformans*	[263,264,265]
Fluconazole, amphotericin B, caspofungin		Chr V monosomy	*C. albicans*	[200,267]
Itraconazole	*pdmA*	Increased copy number	*Aspergillus fumigatus*	[268]
Radicicol	*STI1*, *PDR5*	Increased copy number	*S. cerevisiae*	[194]
Tunicamycin	*ALG7*, *PRE7*, *YBR085C-A*	Increased copy number	*S. cerevisiae*	[199]
4-Nitroquinoline-N-oxide	*ATR1*	Increased copy number	*S. cerevisiae*	[90]
Sterol demethylase inhibitors	Mutated *CYP51*	Increased copy number	*Erysiphe necator*	[269]
Hydroxyurea, caspofungin		Chr II trisomy	*C. albicans*	[200]
Inflammatory host response		Chr V or VI triplication	*C. albicans*	[270]
Modulation of pathogenicity	*Ave1, XLOC_009059*, *XLOC_008951,* etc.	Translocations and inversions	*Verticillium dahliae*	[72]
**Industrially relevant phenotype:**				
Improved growth in maltose-containing medium	*MAL1*, *MAL3*, *MPH2, MPH3*, and *YPR196W*	Increased copy number	*S. cerevisiae*	[86,271]
Improved growth in galactose-containing medium	*GAL7*, *GAL10*, *GAL1*	Increased copy number	*S. cerevisiae*	[187]
Desirable cell aggregation	*FLO1, Lg-FLO1, FLO5 and FLO10*	Increased copy number	*S. cerevisiae*	[184,272]
Low diacetyl production	*ILV5* and *ILV3*	Increased copy number	*S. pastorianus*	[184]
Improved starch consumption	α-amylase gene	Increased copy number	*Aspergillus oryzae*	[273,274]
Increased ethanol yield		Chr XI duplication	*S. cerevisiae*	[28]
Improved xylose fermentation in the presence of ferulic and p-coumaric acids		Chr XIV duplication	*S. cerevisiae*	[275]
**Compensation of mutation:**				
Deletion of *MEC1*	*RNR1*	Increased copy number	*S. cerevisiae*	[276]
Deficiency of *EST2*	*PRP8*, *UTP9*, *KOG1*, and *SCH9*	Chr VIII monosomy	*S. cerevisiae*	[202]
Deletion of *RNR1*	*RNR3*	Increased copy number	*S. cerevisiae*	[206]
Deletion of *RPS24A*	*RPS24B*	Increased copy number	*S. cerevisiae*	[206]
Deletion of *MYO2*	*RLM1*	Increased copy number	*S. cerevisiae*	[91]
Deletion of *JEN1*	*ADY2*	Increased copy number + SNV	*S. cerevisiae*	[277]
Deletion of *GAL7*	*GAL80*	Increased copy number	*S. cerevisiae*	[278]
Deletion of *RGD1*	*NPR2*	Increased copy number	*C. albicans*	[279]

^1^ Tetraploid-specific effect.

## Data Availability

Not applicable.

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
