# Peer review of "The Role of Structural Variation in Adaptation and Evolution of Yeast and Other Fungi"

_genes, 2021, doi:10.3390/genes12050699_

Round 1

Reviewer 1 Report

Authors did great job of summarizing up-to-date knowledge of structural variations in fungi. I wonder if there is some knowledge about SV in higher organisms, for example, in cephalopods?

Line 144: what are "ohnologs"?
Line 217: correct "bene" to been
Line 493: "The loss of chromosome V increased tolerance to antifungals with different mechanisms of action (fluconazole, andamphotericin B, and caspofungin) [200,269], as well as conveyed the ability to utilize L-sorbose as a carbon source due to upregulation of sorbose SOU1 [250 ]." - I am just curious, what was the action behind increased tolerance to antifungals connected with the loss of chromosome V?
Line 524: correct "evert" to ever

Author Response

Dear Reviewer,

Thank you very much for your comments! Please see the point-by-point response below.

Point 1. Authors did great job of summarizing up-to-date knowledge of structural variations in fungi. I wonder if there is some knowledge about SV in higher organisms, for example, in cephalopods?

Response 1: There is a lot of information about SV in higher organisms, e.g. there is a recent Special Issue of Molecular Ecology entirely devoted to this topic (https://onlinelibrary.wiley.com/toc/1365294x/2019/28/6).  Genomic rearrangements are considered to be an important contributor to cephalopod evolution as well (https://www.nature.com/articles/nature14668, https://onlinelibrary.wiley.com/doi/full/10.1002/bies.201900073).

Point 2. Line 144: what are "ohnologs"?

Response 2: Ohnologs are the gene duplicates resulting from a whole genome duplication (not from small-scale duplication events). In the Lines 130-132 of the manuscript we wrote:

“Whereas the S. cerevisiae genome contains roughly 5000 genes, the current genome only contains 1120 pairs of duplicates, approximately half of which belong to the WGD event (so-called “ohnologs”) and the remaining are classified as small-scale duplications [100,101].”

We considered this brief explanation to be sufficient. Do you think that further elaboration is required?

Point 3. Line 217: correct "bene" to been

Response 3: done

Point 4. Line 493: "The loss of chromosome V increased tolerance to antifungals with different mechanisms of action (fluconazole, andamphotericin B, and caspofungin) [200,269], as well as conveyed the ability to utilize L-sorbose as a carbon source due to upregulation of sorbose SOU1 [250 ]." - I am just curious, what was the action behind increased tolerance to antifungals connected with the loss of chromosome V?

Response 4: This question was not fully elucidated and further experiments are required to pinpoint the genes involve in the acquisition of the tolerance to antifungals. However, in the reference [269] it is reported that “monosomy of chromosome 5 causes elevated levels of chitin and repressed levels of 1,3-β-glucan components of the cell wall, as well as diminished cellular ergosterol. Increased deposition of chitin in the cell wall could be explained, at least partially, by a 2-fold downregulation of CHT2 on the monosomic chromosome 5 that encodes chitinase”. Also, it was found that “All 16 monosomic mutants tested acquired resistance to zymolyase” (the cell-wall-degrading enzyme which specifically hydrolyzes 1,3-β bonds in glucan). Thus, it is possible to speculate that the changes in the cell wall and/or membrane permeability for the drugs plays a role in the acquisition of the tolerance.

Point 5. Line 524: correct "evert" to ever

Response 3: done

Reviewer 2 Report

Dear Editor, dear Authors,

I find this review article comprehensive and clearly written. I think though, that the title needs to be reconsidered, as the review centers mainly on the discoveries done in yeasts. The authors indeed mainly mention experiments and examples from Saccaromyces cervisiae and there are only a few citations regarding other fungi. I would therefore make it clearer in the title that this review is centered on yeast and not Fungi in general. Otherwise, I think that the  manuscript deserves publication after a few format mistakes are corrected, I have marked them in the attached PDF revised version.

Author Response

Dear Reviewer,

Thank you very much for your comments! Please see the point-by-point response below.

Point 1. I find this review article comprehensive and clearly written. I think though, that the title needs to be reconsidered, as the review centers mainly on the discoveries done in yeasts. The authors indeed mainly mention experiments and examples from Saccaromyces cervisiae and there are only a few citations regarding other fungi. I would therefore make it clearer in the title that this review is centered on yeast and not Fungi in general. 

Response 1. We fully agree with the Reviewer and suggest the following title: “The Role of Structural Variation in Adaptation and Evolution of Yeast and Other Fungi”.

Point 2. Otherwise, I think that the manuscript deserves publication after a few format mistakes are corrected, I have marked them in the attached PDF revised version.

Response 2. The typos and formatting mistakes were corrected.